# An Optimized Deep Learning Model for Predicting Mild Cognitive Impairment Using Structural MRI

**DOI:** 10.3390/s23125648

**Published:** 2023-06-16

**Authors:** Esraa H. Alyoubi, Kawthar M. Moria, Jamaan S. Alghamdi, Haythum O. Tayeb

**Affiliations:** 1Department of Computer Science, College of Computing and Information Technology, King Abdulaziz University, Jeddah 21589, Saudi Arabia; kmoria@kau.edu.sa; 2Department of Diagnostic Radiology, Faculty of Applied Medical Sciences, King Abdulaziz University, Jeddah 21589, Saudi Arabia; jalghamdi@kau.edu.sa; 3The Neuroscience Research Unit, Faculty of Medicine, King Abdulaziz University, Jeddah 21589, Saudi Arabia; hostayeb@kau.edu.sa

**Keywords:** mild cognitive impairments, deep learning, entorhinal cortex, magnetic resonance imaging, transfer learning

## Abstract

Early diagnosis of mild cognitive impairment (MCI) with magnetic resonance imaging (MRI) has been shown to positively affect patients’ lives. To save time and costs associated with clinical investigation, deep learning approaches have been used widely to predict MCI. This study proposes optimized deep learning models for differentiating between MCI and normal control samples. In previous studies, the hippocampus region located in the brain is used extensively to diagnose MCI. The entorhinal cortex is a promising area for diagnosing MCI since severe atrophy is observed when diagnosing the disease before the shrinkage of the hippocampus. Due to the small size of the entorhinal cortex area relative to the hippocampus, limited research has been conducted on the entorhinal cortex brain region for predicting MCI. This study involves the construction of a dataset containing only the entorhinal cortex area to implement the classification system. To extract the features of the entorhinal cortex area, three different neural network architectures are optimized independently: VGG16, Inception-V3, and ResNet50. The best outcomes were achieved utilizing the convolution neural network classifier and the Inception-V3 architecture for feature extraction, with accuracy, sensitivity, specificity, and area under the curve scores of 70%, 90%, 54%, and 69%, respectively. Furthermore, the model has an acceptable balance between precision and recall, achieving an F1 score of 73%. The results of this study validate the effectiveness of our approach in predicting MCI and may contribute to diagnosing MCI through MRI.

## 1. Introduction

A computer-aided diagnosis system (CADS) helps to automate the diagnosis process of diseases. Clinicians are increasingly using CADS to help them detect and interpret diseases. CADS identifies regions of an image that may reveal certain problems and notifies doctors during image interpretation [1]. Typically, CADS includes pre-processing, feature extraction, and classification [2].

Alzheimer’s disease (AD) is a neurological disease most commonly linked to memory and cognitive loss. Neurodegenerative disorders are not curable [3]. The goal of medicine is to enhance patients’ well-being and slow the progression of the disease. However, early diagnosis may provide AD patients with better treatment outcomes than patients who discover the disease late. Mild cognitive impairment (MCI) disease is an initial stage of AD. MCI is a form of memory loss or a decline in cognitive skills, such as language or vision. Patients with MCI go through a stage in which they have cognitive deficiencies that are not severe enough to cause dementia. MCI is a condition in which patients experience greater memory or thinking difficulties than people of the same age. Studies show that individuals with MCI are more likely than people with normal cognitive abilities to develop Alzheimer’s disease within a few years [4]. According to the National Institute on Aging [5], dementia is estimated to develop in 10% to 20% of MCI patients aged 65 or older within one year.

Brain imaging can detect MCI before clinical symptoms appear [6]. Researchers use medical imaging modalities to diagnose AD and MCI, including positron emission tomography (PET) and magnetic resonance imaging (MRI) scans. These studies classify MCI disease based on its biomarkers (indicators that the disease exists), such as decreased grey matter volume. Figure 1 illustrates how the grey matter volume in an AD patient’s brain is less than that in a healthy brain, whereas the grey matter volume for the MCI patient is less than that of the healthy brain but more than that of the brain affected by AD.

Another significant biomarker related to MCI is the shrinkage of the hippocampus and entorhinal cortex (EC) area. In contrast with the hippocampus, the EC area is important in the early detection of MCI as studies have demonstrated that the EC area shrinks before the hippocampus region [8]. However, there are limited studies exploring the EC area for the diagnosis of MCI as EC is challenging to analyze due to its small size, which makes it difficult to detect with the human eye.

Recently, machine learning approaches are being more widely used to automatically classify medical images [9,10,11]. However, the feature engineering step in machine learning approaches is time-consuming and requires expert knowledge. In contrast, deep learning models can automatically learn relevant features from the raw data, eliminating the need for explicit feature engineering. A recent study [12] employed a mathematical approach called ellipse fitting to detect abnormalities in medical images. However, the ellipse fitting algorithm is sensitive to outlier data when the organ being analyzed has an irregular shape, which can significantly affect the algorithm’s accuracy.

Significant advancements have been made in the field of brain tumor detection. Deep learning algorithms have been used to analyze brain images and identify patterns of brain tumors. Recent studies have demonstrated that convolutional neural networks (CNNs) are highly accurate at detecting brain tumors [13,14,15]. In addition, deep learning algorithms have been shown to distinguish between various forms of brain tumors, assisting medical professionals in determining the best strategy for treatment [16].

Inception architecture has demonstrated promising outcomes in brain studies, achieving notable performance in brain tumor detection [17,18]. Several advanced techniques were adopted in inception architecture, including factorized convolution and auxiliary classifiers. In factorized convolution, the number of parameters in the network is reduced by factorizing the convolutional filters into two smaller filters, minimizing the overall complexity of the model. Auxiliary classifiers refer to classifiers that are added to the network at intermediate levels to avoid overfitting and enhance prediction accuracy.

Support vector machine (SVM) is a supervised classification algorithm that separates the classification points by hyperplane (decision boundary) [19]. The maximum distance between the hyperplane and the closest point is called the margin; this is what gives the SVM its robustness, meaning that it is dependable and avoids errors as much as possible. In practice, the arguments passed while creating the SVM classifier strongly influence the model outcome. Kernels, gamma, and C are augments that must be tuned to achieve the highest accuracy possible. The combination of a CNN as a feature extractor and an SVM as a classifier has been shown to perform medical imaging tasks effectively [20].

Using pre-trained CNN architectures (such as VGG, Inception, and ResNet) to extract features of the EC is a powerful technique that can save time and resources compared to training a model from scratch. The effectiveness of this technique can be explained theoretically in the following ways: (1) using the transfer learning technique as a feature extractor can capture both low-level and high-level visual information of the EC area because these models have been trained on large datasets, such as ImageNet, with 10,000 classes. (2) By using our proposed method of extracting the features of the EC area, we reduce the chance of overfitting issues that can occur since training a CNN from scratch on a small dataset can lead to overfitting.

In this study, we aim to perform automatic classification of MCI disease from MRIs using the EC area. To achieve this goal, we define the study objectives as follows: (1) construct a dataset for the EC area from MRIs with normal and MCI subjects; (2) investigate using different collections of MRI slices as inputs for the classification system; (3) explore different neural network architectures, including VGG16, Inception-V3, and ResNet50, to extract the features of the EC area; and (4) classify subjects using machine learning algorithms, including CNN and SVM. There are two important contributions of this study. First, this work expands on the studies conducted on the EC area since there is a limited body of knowledge addressing EC and linking it with neurological diseases. Second, to our knowledge, no dataset for the EC area is available. In this work, we used MRIs from the Alzheimer’s Disease Neuroimaging Initiative (ADNI) dataset to extract the left and right EC areas and use them as inputs for the proposed classification models.

The remaining sections of this document are organized as follows: Section 2 presents the state-of-the-art classification systems for predicting MCI, Section 3 explains the proposed system, Section 4 reports the experimental results, and Section 5 presents the conclusions and recommendations for further studies.

## 2. Related Work

Many studies have been conducted to classify MCI and normal control (NC) samples, some of which used the whole brain to diagnose MCI. For this literature review, we focused on the studies that were conducted based on the region of interest (ROI) approach as they relate more closely to the purpose of this study. The ROI approach considers only the brain’s informative part because including the whole brain will also involve some areas of the brain that are unaffected by the disease [21]. This approach reduces the overall complexity of the system by reducing the inputs. For these studies, the ROI was either the hippocampus region or the entorhinal cortex area. The related works are divided into three sections based on the inputs to the model: the whole brain, the hippocampus region, and the EC area.

### 2.1. Classification Systems Based on the Whole Brain Image

Mehmood et al. [22] proposed a deep learning system for classifying three groups: NC, late MCI (LMCI), and early MCI (EMCI). The authors used the grey matter biomarker to differentiate between the samples, employing the VGG-19 network architecture. The convolution layer was fixed, while the last two fully connected layers and the classification layers were modified. The authors proposed two models: Model 1 freezes the first eight convolutional layers with three max pooling layers, and Model 2 freezes the first twelve convolutional layers with four max pooling layers. Model 2 achieved a higher result compared to Model 1. The highest accuracy obtained was 87% for differentiating between NC and EMC, and an accuracy of 89% for classifying NC vs. LMCI. Different reports used the transfer learning approach to train the systems to deal with limited data [22,23], producing comparable classification performance. However, the computation requirement was enormous since the whole brain MRIs are used as inputs.

Senanayake Upul et al. [24] combined two modalities: MRI and neuropsychological measures with 35 features to diagnose MCI; however, the neuropsychological features are not presented clearly in the paper. The obtained accuracy was 75% for classifying MCI samples from NC. Other previous studies removed the skull and the neck from the MRI scans [25,26,27]. Wang et al. [26] intensity normalization was also performed on the MRIs before they were fed into the CNN model. The authors achieved an accuracy of 86%. However, the dataset used to validate the system was relatively small. Table 1 lists the parameters used in previous studies conducted on whole brain images.

### 2.2. Classification Systems Based on the Hippocampus Region

The hippocampus is a well-known brain area known to be damaged by MCI [29]. The hippocampus area is extensively studied for MCI and AD diagnosis as presented in those studies [30,31,32,33].

In 2017, Aderghal et al. [30] implemented an MCI diagnosis system using only the hippocampus region. The authors used an automated anatomical labeling (AAL) atlas for extracting the hippocampus area, achieving an accuracy for the diagnosis system of 66%. One year later, Aderghal et al. [31] built a model that combines MRI and mean diffusivity (MD) imaging modalities and applied it to the hippocampus region. To prepare the MRI to enter the CNN, they registered all the images in the same coordinate system, the Montreal Neurological Institute (MNI) space. MNI is a three-dimensional coordinate system that maps brain structures onto a standard template brain; the hippocampal region is then selected using the AAL atlas. The accuracy of the implemented model was 80% for classifying MCI vs. NC.

Choi et al. [32] automatically extracted the hippocampus area using 3D slicer software https://www.slicer.org/, (accessed on 20 November 2022), while CNN was used for feature extraction and classification of the hippocampus. The achieved accuracy for the model was 78.1% for MCI vs. NC.

Depending on the texture and morphometric characteristics of the hippocampus region, Madusanka et al. [33] proposed a model for classifying MCI and NC from MRIs. The authors used 50 samples for each class. The maximum accuracy obtained was 82% for classifying between the samples using the texture features. Table 2 lists the architectures used for the classification of MCI using the hippocampus region.

### 2.3. Classification Systems Based on the Entorhinal Cortex Region

According to [34], the EC volumes drastically decrease in AD patients compared to those with MCI, and the volume of EC is less in MCI patients compared to cognitively normal people. This indicates that the EC area is an important biomarker for the diagnosis of AD and MCI. However, limited studies have addressed the use of the EC for MCI diagnosis.

Leandrou et al. [35] implemented a machine learning classifier, binary logistic regression, for the diagnosis of MCI samples. The model’s input is MRIs that include only the EC region; the methodology is based on the texture feature of the entorhinal cortex. The authors found that the EC’s textural features significantly differed between the NC and MCI, with a classification accuracy of 71%.

Devivo et al. [36] implemented a logistic regression model to differentiate between NC and MCI. The method was based on the volume of the hippocampus, EC, and other cortical regions segmented in the MRI scans. The study used two datasets, ADNI and Boston University Alzheimer’s Disease Center (BU-ADC), obtaining an accuracy of 93.75%.

Li et al. [37] measured the EC morphology with the objective to show how EC volume, thickness, and surface area differ among three groups: AD, amnestic mild cognitive impairment (aMCI), and NC. MCI-s is the single domain deficiency of MCI, including language, visuals, and attention, whereas MCI-m are the multiple domains for MCI [38]. For the segmentation process, the automatic segmentation of the EC was performed using FreeSurfer. The Desikan–Killiany atlas was used to drive the left and right EC volume measures. The classification process is based on a covariance analysis (ANCOVA) statistical approach. The main finding of the study was that, when EC thickness and volume were combined, the area under the curve (AUC) value was larger than when the thickness was used alone. Table 3 lists the classification systems that use EC for the diagnosis of MCI.

Previous studies on the EC area have been conducted using relatively small datasets (as shown in Table 3), which can limit the generalizability of results and increase the risk of overfitting. Moreover, deep learning algorithms have not been extensively investigated to extract the features and identify the specific pattern of the EC area. Further research is needed to evaluate the accuracy of deep learning algorithms in diagnosing MCI disease.

## 3. Proposed System

In this study, optimized deep learning systems are implemented to differentiate between MCI and NC subjects. The proposed classification system includes the following phases: MRI pre-processing, extraction of the EC area, feature extraction, and classification.

For the pre-processing phase, the first step is to align all MRI images to a standard template, the International Consortium for Brain Mapping (ICBM152). Second, the MRIs are prepared by performing skull stripping, motion correction, intensity normalization, cortical service segmentation, and parcellation. To extract the EC area, the Desikan–Killiany atlas is used. Next, the VGG16, Inception-V3, and ResNet50 networks were used as feature extraction for the EC area. Subsequently, the classification process is applied using two different classifiers: CNN and SVM. Figure 2 depicts the workflow of the proposed prediction system.

### 3.1. Constructing the Dataset

To our knowledge, there is no public dataset available with segmented EC to perform the experiments. For this reason, MRIs from the ADNI dataset [39] were used to segment the EC and construct the ground truth.

The size of the dataset is 779 3D MRIs, including MCI and NC MRIs. The number of participants included in this study is 188 participants, of whom 95 are NC and 93 have MCI. Each participant has one or more MRIs. The total number of MRIs for NC is 442, while the number of MRIs for the MCI is 337. The participants range in age from 55 to 90 years old. Table 4 provides demographic information for the dataset included in this study.

The data included in this study are T1-weighted MRIs. The field strength of the MRIs is 3 Tesla, and the 3D images are in Neuroimaging Informatics Technology Initiative (NIfTI) format. The data used are in the original format, meaning no pre-processing was performed previously by the dataset provider to ensure total control over the data. The dimensions of each 3D MRI are (256, 256, 170).

### 3.2. Pre-Processing

The first step in pre-processing is performing the MRI alignment. The objective of the alignment process is to center all the images such that, when a specific area is selected, the same area will be used for all the images. This step is crucial as not all images have the same orientation or centering due to the manual screening completed by the physicians. For example, the skull stripping process could be affected negatively if the images are not centered. This could result in the tool extracting a part from the skull because it does not differentiate between the skull and the brain. Moreover, alignment can also solve the problem when the head is not oriented and tilted forward or backward.

Typically, this is completed by linearly aligning the image to a standard template, ICBM152. In this work, we used the statistical parametric mapping (SPM12) tool [40] to perform the alignment process for all the MRIs.

Further, 3D MRI volumes in NIfTI format were used to construct the 2D surface to perform the analysis. FreeSurfer is an open-source tool that is publicly available online [41]. The preprocessing approach includes thirty-one steps for all processes to complete, which are grouped into three phases: Autorecon 1, Autorecon 2, and Autorecon 3. The output of each phase is the input for the subsequent phase, as shown in Figure 3. The first five steps in Autorecon 1 address the volume itself, skull stripping, motion correction, and intensity normalization. These steps are followed by Autorecon 2 for performing the segmentation for the cortical surface. The last phase, Autorecon 3, generates the statistical data and cortical parcellation.

### 3.3. Extracting the Entorhinal Cortex Area

To automatically extract the EC area, the Desikan–Killiany atlas was used. The brain regions are defined by an anatomical atlas. Various atlases are available, such as the Desikan–Killiany and Destrieux atlases; each atlas has different brain regions, which are segmented and painted with distinct colors. In this study, we used the Desikan–Killiany atlas because it contains the anatomical location of the EC area and uses a dataset of 40 MRI images for labeling ROIs in the left and right hemispheres [43]. Figure 4 lists the names of the regions that exist in the Desikan–Killiany atlas, along with the axial, sagittal, and coronal views of those regions. The EC area painted in the figure in red color code 1006 represents the left EC, while the area painted in 2006 represents the right EC.

The extracted 3D images are in the shape (256, 256, 256). Figure 5 shows samples of the extracted left and right EC areas in this study for participants with and without MCI disease. The images show how the EC is significantly different in size and shape for the two groups. The images for participants without MCI show a normal size of the EC area, while the EC area for participants with MCI appears smaller in size.

### 3.4. Feature Extraction and Classification

The first step in this section is performing the feature extraction for the segmented EC images resulting from the previous step. This study uses the transfer learning concept for feature extraction of the EC area to improve the detection of features. The convolutional base (convolutional and pooling layers) of the previously trained models is used. Three CNN models are examined: VGG16, Inception-V3, and ResNet50. Figure 6 illustrates the six experiments conducted in this study.

VGG16 [44] consists of sixteen weight layers, thirteen convolutional layers, five max pooling layers (no trainable weights), and three dense layers.

Inception-V3 [45] is the second pre-trained model used. The main concept of inception architecture is going wider instead of going deeper. Inception- V3 consists of forty-eight layers; the block of the inception module consists of three convolutional layers and one max pooling concatenated together.

ResNet50 [46] has fifty layers, forty-eight convolutional layers, one max pooling layer, and one average pooling layer.

The second step after the feature extraction is classification. In this approach, two different classification methods are used that are based on binary classification (NC and MCI). The last fully connected layer is removed as it is responsible for multiclassification, which is not used in this approach.

The extracted features are in 3D form, so we flatten them before they run through the classifier. We examined two different classifiers for comparison purposes: the first classifier is a CNN layer with a sigmoid activation function to perform binary classification tasks (NC vs. MCI); the second is the SVM classifier. The 5-fold cross-validation technique is used for the resampling of the EC images. The data are divided into five folds; the first fold is used as a testing set, and the remaining 4 sets are used for training the model. The second fold uses the second set as a test set and the remaining 4 sets as training sets. This procedure is repeated for each of the five groups. The final result is the mean for all classification results obtained for each group.

## 4. Experimental Design and Results

The experiments conducted to validate the proposed methodology for the prediction of MCI disease are described in detail in this section. The evaluation matrices adopted to evaluate the performance of each experiment are introduced. Moreover, the experiments conducted to produce an efficient classification system capable of differentiating between MCI and NC samples are discussed. The experiments mainly focus on the brain slices used as inputs for the classification system, the feature extraction techniques, and the classifier. The following subsections clarify the objective of each experiment along with the results obtained.

Various metrics were used to evaluate the performance of each experiment, including accuracy, F1 score, sensitivity, specificity, and AUC [47]. For these evaluation matrices, a score of 1 indicates that all samples were predicted correctly, while a score of 0 indicates that none of the samples were predicted correctly. The following list provides the definitions and equations for each matrix used.

Accuracy measures the ratio of the correctly classified samples to the total of samples in the dataset.
(1)$$\mathrm{Accuracy}=\frac{\left(\mathrm{True}\mathrm{Positive}+\mathrm{True}\mathrm{Negative}\right)}{\left(\mathrm{True}\mathrm{Positive}+\mathrm{True}\mathrm{Negative}+\mathrm{False}\mathrm{Positive}+\mathrm{False}\mathrm{Negative}\right)}$$F1 score measures the harmonic mean of the precision and recall. The definitions and matrices for precision and recall are presented below.
(2)$$F1\mathrm{Score}=\frac{2\times (\mathrm{Precision}*\mathrm{Recall})}{\left(\mathrm{Precision}+\mathrm{Recall}\right)}$$-Precision measures the ratio of the correctly classified samples to all samples assigned to that class.
(3)$$\mathrm{Precision}=\frac{\left(\mathrm{True}\mathrm{Positive}\right)}{\left(\mathrm{True}\mathrm{Positive}+\mathrm{False}\mathrm{Positive}\right)}$$-Recall, also known as Sensitivity or true positive rate, measures the proportion of actual positive cases that are correctly identified by a classifier.
(4)$$\mathrm{Recall}=\frac{\left(\mathrm{True}\mathrm{Positive}\right)}{\left(\mathrm{True}\mathrm{Positive}+\mathrm{False}\mathrm{Negative}\right)}$$Specificity, also known as true negative rate, measures the ratio of the correctly classified negative samples to the total of negative samples in the dataset.
(5)$$\mathrm{Specificity}=\frac{\left(\mathrm{True}\mathrm{Negative}\right)}{\left(\mathrm{True}\mathrm{Negative}+\mathrm{False}\mathrm{Positive}\right)}$$Area under the curve (AUC) is calculated using the Receiver Operating Characteristic (ROC) curve, which is a graphical representation of the performance of a binary classification model. The ROC curve is a plot of the true positive rate against the false positive rate for the classifier at different threshold values. The AUC represents the overall performance of the classifier across all possible threshold values.
(6)AUC=∫[0,1]TPR(FPR)dFPR

### 4.1. Experiment 1: Investigate the Use of Different Collections of MRI Slices as Inputs for the Classification System

The original shape of the EC area after the extraction from the Desikan–Killiany atlas is (256, 256, 256) for each MRI. Each 3D image is transformed into 2D images for visualization purposes, resulting in 256 images in the shape of (256, 256) for each MRI. After exploring the images, we observe that some images are uninformative for classifying the MCI since the EC area is either incomplete or cannot be observed in the slice, as shown in Figure 7. To obtain the highest accuracy for the classification model, various experiments were implemented using different groups of 2D MRI slices. The feature extraction network and the classifier were fixed for each experiment to isolate the influence of the MRI slice groups on the accuracy. The original size of the images was 256, 256 pixels; the new size of the images after the cropping process was 100, 70 pixels. The hyperparameters used for training the models included an epoch value of 50 and the adaptive moment estimation (Adam) optimization method. Detailed information for each scenario is listed below.

Experiment 1.1: Include all MRI slices (slice 0–slice 255)In the first experiment, we included all the MRI slices as inputs for the CNN classification system.Experiment 1.2: Select the range of the MRI slices (slice 130–slice 140)As discussed above, some slices are not useful for the classification process. For the second experiment, slices 130–140 are included. This range of slices is chosen based on the visualization of the images. From these slices, we observed that the EC area was clearer and more informative than for other MRI slices, as shown in Figure 7.Experiment 1.3: Exclude uninformative MRI slices.Rather than choosing a specific range of MRI slices, in this experiment, the MRI slices with the black pixels (empty slices) are deleted as they do not provide information for the classification task. The pillow library in Python is used to analyze each MRI slice; if all the pixels of the slice are black, the slice is ignored.

Experiment 1 Results: Table 5 shows the accuracy obtained using different groups of MRI slices. By excluding the uninformative slices (Experiment 1.3), we obtained the highest accuracy of 60% compared to other experiments. The lowest accuracy was gained by including all 256 MRI slices in the classification model. Based on this result, all subsequent experiments were conducted using the dataset used in Experiment 1.3 as it achieved the highest accuracy.

### 4.2. Experiment 2: Investigate Implementing the PCA Technique for Feature Reduction

Many features are noisy, cause overfitting, and slow down the training and testing models. As the number of features extracted from the EC area is very large (approximately 51,200 features), reducing the number of features is needed before using the SVM classifier. In this experiment, we used principal component analysis (PCA) to reduce the number of features included in the model, with the objective of identifying the minimum number of features that provides the maximum accuracy possible.

The optimal number of principal components (PCs) is obtained with maximum variance. Based on the Scree plot shown in Figure 8, 2000 PCs are required to explain approximately 95% of the variance, which is significantly less than the 51,200 features. Different values of PCs are tested in this experiment to see how the accuracy responds. Table 6 shows the number of PCs used and the corresponding accuracy. We observe that the accuracy was 45% when the number of PCs was 2500, and the accuracy increased as the number of PCs increased. At approximately 10,000 PCs, the accuracy started to decrease. The maximum accuracy of 53% was observed for 7500 PCs.

### 4.3. Experiment 3: Evaluate the Accuracy of the SVM Classifiers with Tuned Parameters

For tuning the parameters of the SVM classifier, a grid search is used. We examined different values for the four key parameters of the SVM classifier: kernel, C, gamma, and degree. In this experiment, the model inputs and the feature extraction network remain fixed. The highest accuracy of 56% was obtained using the parameters listed in Table 7. Below is a description of each hyperparameter tuned in this experiment.

For the kernel hyperparameter, three types of kernel were tested: linear, radial basis function (rbf), and poly. Both poly and rbf produce the highest accuracy, 56%. The poly kernel with degree of 2 is chosen because the execution time is less than that of rbf.

The C hyperparameter is responsible for adding a penalty for incorrectly classified points. The C values examined in this experiment are [0.1, 1, 10, and 100]. Note that increasing the value of C can result in overfitting and poor generalization for testing datasets.

Smaller gamma values are known to produce a generalized decision boundary, whereas larger gamma values produce a complex decision boundary that may overfit the training data. Sklearn library [48] provides two arguments for gamma: ‘scale’ and ‘auto’. The scale value represents 1/(n_features X.var()). The n_features is the number of features in the dataset, whereas the X.var expresses the variance. For the ‘auto’ argument, the value set for the gamma is a fixed number based on the number of features. Both arguments were tested in this experiment, with a higher accuracy value obtained for the scale argument.

### 4.4. Experiment 4: Tuning VGG16, Inception-V3, and ResNet50 Network Parameters

In this experiment, our objective is to improve the performance of the CNN classifier by tuning the parameters of each pre-trained model (VGG16, Inception-V3, and ResNet50) separately. In the following tests, we examined different values of the epoch size, optimization method, and learning rate for each model.

Tuning Parameters of the VGG16Epochs: The performance of the VGG16 based on different epoch values is evaluated in this experiment. The range for the examined epoch values was 25 to 300 epochs, as listed in Table 8. As can be seen in the table, the accuracy remains nearly constant (within 54%) from epochs 25 to 200, whereas the accuracy increases at 300 epochs.sensors-23-05648-t008_Table 8Table 8Accuracy of VGG16 model across different values of epoch parameter.ClassifierFeature ExtractionEpochAccuracyCNNVGG16250.55CNNVGG16500.55CNNVGG161000.54CNNVGG162000.54CNNVGG163000.57Optimization Method: The different optimization methods evaluated in this test are listed in Table 9. The model is tuned with Adam, stochastic gradient descent (SGD), and root mean square propagation (RMsprop). The highest accuracy was obtained with the SGD optimizer, while the lowest accuracy was obtained with the RMSprop optimizer.
sensors-23-05648-t009_Table 9Table 9Accuracy of the VGG16 model for different optimization methods.ClassifierFeature ExtractionEpochOptimization MethodAccuracyCNNVGG16300Adam0.57CNNVGG16300SGD0.60CNNVGG16300RMSprop0.54Learning rate: The model performance is evaluated for different learning rate values, as shown in Table 10. The range of learning rate values is 0.1–1 × 10−5. The optimizer used is SGD, and the epoch size is 300. The resulting accuracy obtained with a large learning rate (0.1) is 62%, while the accuracy with a small learning rate (1 × 10−5) decreases to 59%, and the accuracy for the middle learning rate values (0.01 and 0.001) increases to 63%.
sensors-23-05648-t010_Table 10Table 10Accuracy of the VGG16 model for different values of the learning rate parameter.ClassifierFeature ExtractionEpochOptimization MethodLearning RateAccuracyCNNVGG16300SGD0.10.62CNNVGG16300SGD0.010.63CNNVGG16300SGD0.0010.63CNNVGG16300SGD0.00010.60CNNVGG16300SGD0.000010.59Tuning Parameters of Inception-V3Epochs: The performance of the Inception-V3 based on different epoch values is evaluated in this experiment. The range for the examined epoch values was 25–300 epochs, as listed in Table 11. The highest accuracy obtained is 67% for 200 epochs.
sensors-23-05648-t011_Table 11Table 11Accuracy of Inception-V3 model for different epoch values.ClassifierFeature ExtractionEpochAccuracyCNNInception-V3250.62CNNInception-V3500.63CNNInception-V31000.65CNNInception-V32000.67CNNInception-V33000.64Optimization Method: The different optimization methods evaluated in this test are listed in Table 12. The model is tuned with Adam, SGD, and RMsprop. The highest accuracy was obtained using the Adam optimizer, with an accuracy of 67%, while the lowest accuracy of 49% was obtained with the RMSprop optimizer.
sensors-23-05648-t012_Table 12Table 12Accuracy of Inception-V3 for different optimization methods.ClassifierFeature ExtractionEpochOptimization MethodAccuracyCNNInception-V3300Adam0.67CNNInception-V3300SGD0.56CNNInception-V3300RMSprop0.49Learning rate: The model performance is evaluated for different learning rate values, as shown in Table 13. The range of learning rate values is 0.1–1 × 10−5. The optimizer used is Adam, and the epoch size is 200. The resulting accuracy obtained with a large learning rate (0.1) is 60%, while the accuracy with a small learning rate (1 × 10−5) is 61%, and the highest accuracy of 67% was obtained with a learning rate of 0.0001.
sensors-23-05648-t013_Table 13Table 13Accuracy of Inception-V3 for different learning rate values.ClassifierFeature ExtractionEpochOptimization MethodLearning RateAccuracyCNNInception-V3300SGD0.10.60CNNInception-V3300SGD0.010.66CNNInception-V3300SGD0.0010.63CNNInception-V3300SGD0.00010.67CNNInception-V3300SGD0.000010.61Tuning Parameters of ResNet50Epochs: The performance of the ResNet50 based on different epoch values is evaluated in this experiment. The range of examined epoch values is 25–300 epochs, as listed in Table 14. The highest accuracy achieved is 63% for both 100 and 300 epochs.
sensors-23-05648-t014_Table 14Table 14Accuracy of the ResNet50 model for different epoch values.ClassifierFeature ExtractionEpochAccuracyCNNResNet50250.54CNNResNet50500.61CNNResNet501000.63CNNResNet502000.61CNNResNet503000.63Optimization Method: The different optimization methods evaluated in this test are listed in Table 15. The model is tuned with Adam, SGD, and RMsprop. The highest accuracy was obtained with the RMSprop optimizer, while the lowest accuracy was obtained with the SGD optimizer.
sensors-23-05648-t015_Table 15Table 15Accuracy of the ResNet50 for different optimization methods.ClassifierFeature ExtractionEpochOptimization MethodAccuracyCNNResNet50300Adam0.63CNNResNet50300SGD0.55CNNResNet50300RMSprop0.65Learning rate: The model performance is evaluated for different learning rate values, ranging from 0.1 to 1 × 10−6. The optimizer used is RMSprop, and the epoch size is 300. The highest accuracy value of 66% was obtained at learning rates of 0.01 and 1 × 10−5, as highlighted in Table 16.

**Table 16 sensors-23-05648-t016:** Accuracy of the ResNet50 for different learning rate values.

Classifier	Feature Extraction	Epoch	Optimization Method	Learning Rate	Accuracy
CNN	ResNet50	300	RMSprop	0.1	0.61
CNN	ResNet50	300	RMSprop	0.01	0.66
CNN	ResNet50	300	RMSprop	0.001	0.64
CNN	ResNet50	300	RMSprop	0.0001	0.65
CNN	ResNet50	300	RMSprop	0.00001	0.66
CNN	ResNet50	300	RMSprop	0.000001	0.61

### 4.5. Experiment 5: Choose the Optimal Combinations of Feature Extraction Techniques and the Classifier

This experiment examines different combinations of feature extraction networks tuned in previous experiments along with different classifiers. Matrices used to evaluate the results include accuracy, F1 score, sensitivity, specificity, and AUC.

Experiment 5.1: VGG16 + CNNFor this experiment, the VGG16 network architecture was employed for feature extraction, while CNN was used to classify MCI and CN samples. The model correctly predicted 70% of the dataset samples. Moreover, the model showed an acceptable balance between precision and recall, achieving an F1 score of 66%. The model correctly identified 69% of the positive cases in the dataset and 70% of the negative samples. The obtained AUC value was 68%.Experiment 5.2: Inception-V3 + CNNIn this experiment, the Inception-V3 network architecture was used for feature extraction, while the CNN was used to classify MCI and CN samples. The model correctly predicted 70% of the dataset samples and demonstrated a good balance between precision and recall, as evidenced by its F1 score of 73%. The model correctly identified 90% of the positive cases and 54% of the negative samples in the dataset. The obtained AUC value was 69%.Experiment 5.3: ResNet50 + CNNThe ResNet50 network architecture was used for feature extraction, while the CNN was used to classify MCI and CN samples. The model correctly predicted 73% of the dataset samples and demonstrated an acceptable balance between precision and recall, as evidenced by its F1 score of 65%. The model correctly identified 58% of the positive cases and 84% of the negative samples in the dataset. The obtained AUC value was 63%.Experiment 5.4: VGG16 + SVMThe VGG16 network architecture was used for feature extraction, while the SVM classifier was used to classify MCI and CN samples. The model correctly predicted 76% of the dataset samples, which was the highest accuracy value achieved compared to the other five experiments. Furthermore, the model achieved an F1 score of 47%. The model correctly recognized 34% of the positive cases in the dataset and 64% of the negative samples. The obtained AUC value was 63%.Experiment 5.5: Inception-V3 + SVMThe Inception-V3 network architecture was used for feature extraction, while the CNN was used to classify MCI and CN samples. The model correctly predicted 66% of the dataset samples and achieved an F1 score of 47%. The model also correctly identified 66% of the positive cases and 45% of the negative samples in the dataset. The obtained AUC value was 63%.Experiment:5.6: ResNet50 + SVMThe ResNest50 architecture was used for feature extraction, while the SVM classifier was used to classify MCI and CN samples. The model correctly predicted 69% of the dataset samples and achieved an F1 score of 58%. The model correctly identified 48% of the positive cases in the dataset and 68% of the negative samples. The obtained AUC value was 67%.

Table 17 shows the results of the classification of MCI vs CN samples. The performance matrices for the six experiments conducted in this study are visualized in Figure 9.

In this work, we ran FreeSurfer on an Aziz supercomputer [49]. The processes performed by FreeSurfer are resource-intensive, requiring vast quantities of CPU time, memory, and disk space. We processed the MRI data in parallel, which helped to reduce the time needed to extract the EC data. All the data used in this work were processed on the same machine using the Aziz supercomputer, whereas the deep learning experiments were implemented on Google Colaboratory Pro. The device used to implement the experiments was an Intel i7-1165G7 with 16 GB of RAM.

### 4.6. Experiment 6: Comparison with State-of-the-Art MCI Classification Systems

Leandrou et al. [35] used the EC area for the classification of MCI vs NC and they validated their approach on the ADNI dataset using a machine learning classifier implemented with binary logistic regression. The model’s input was MRIs that include only the EC region. The texture feature of the EC was used to differentiate between the samples. Their reported AUC value was 71%. In this study, the two models using the Inception-v3 and ResNet50 as feature extraction and CNN as classifier achieved 69% for AUCs, which is comparable to the results obtained by Leandrou et al. [35]. These results demonstrate that the proposed method was able to differentiate between NC and MCI samples using the EC area.

## 5. Conclusions

This study implements a deep learning system for predicting MCI using the EC area as a biomarker. Investigating the EC area is crucial as the change in this area occurs before the hippocampus, which will help in the early diagnosis of MCI. In addition, limited studies have used EC as a biomarker for MCI because the area is small compared with the hippocampus area; thus, it is a challenge to detect the changes. The approach in this study uses the EC area to predict MCI using neural networks and machine learning algorithms.

Experiments in this research were conducted to produce an efficient classification system capable of differentiating between MCI and NC samples. The experiments mainly focused on brain slices used as inputs for the classification system, the feature extraction techniques, and the classifier. We performed investigations using different groups of MRI slices as inputs for the classification system. By excluding the uninformative slices, we obtained the highest accuracy compared to other experiments. We also investigated implementing the PCA technique for feature reduction before implementing the SVM classifier. We obtained a maximum accuracy of 53% with 7500 PCs. In this research, we evaluated the accuracy of the SVM classifier with tuned hyperparameters for kernel, C, gamma, and degree. The highest accuracy of the model was obtained for C = 0.1, degree = 2, gamma = scale, and kernel = poly.

We improved the performance of the CNN classifier by separately tuning the parameters of each pre-trained model (VGG16, Inception-V3, and ResNet50). We also examined different values for the epoch size, optimization method, and learning rate for each model. We found that using Inception-V3 as a feature extractor and CNN as a classifier produced the highest performance compared to the other models implemented.

A limitation of this research is that the model inputs are limited to MRI data, whereas other types of data, such as clinical, genetic, and genomics, are considered to be out of scope. In future work, we will extract the features of the hippocampus and EC area and use them as inputs for the proposed classification system. Combining the hippocampus with the EC area could improve the performance of the classification system.

## Figures and Tables

**Figure 1 sensors-23-05648-f001:**
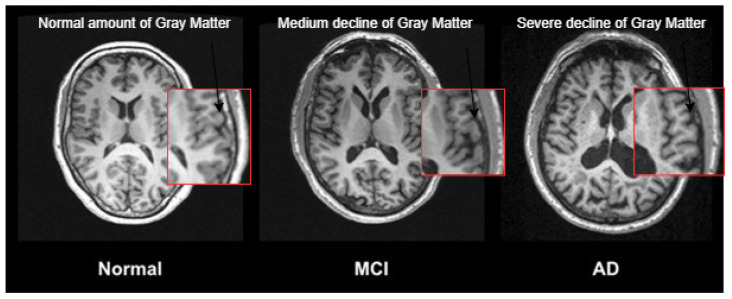
T1-weighted MRI shows the volume of grey matter for normal, MCI, and AD subjects [7]; the normal amount of grey matter in left subject, whereas the grey matter decreases gradually from the MCI to the AD subjects.

**Figure 2 sensors-23-05648-f002:**
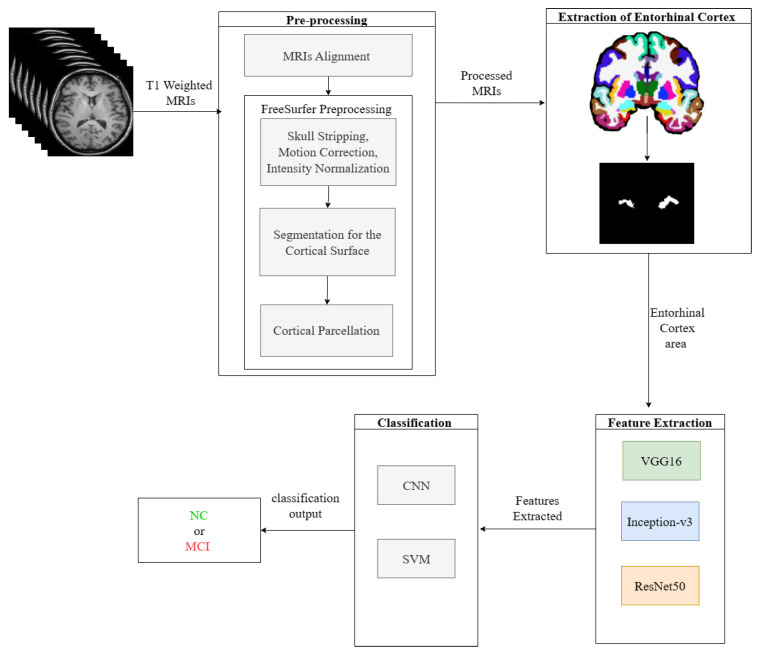
Proposed MCI prediction system.

**Figure 3 sensors-23-05648-f003:**
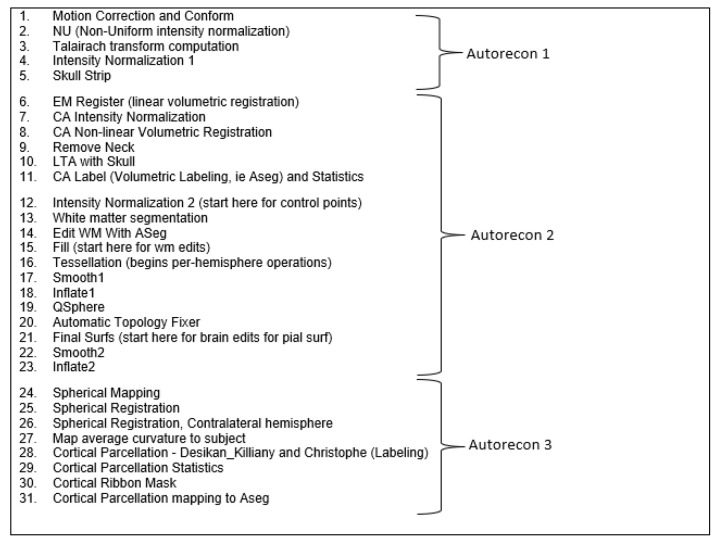
Autorecon processing stages [42].

**Figure 4 sensors-23-05648-f004:**
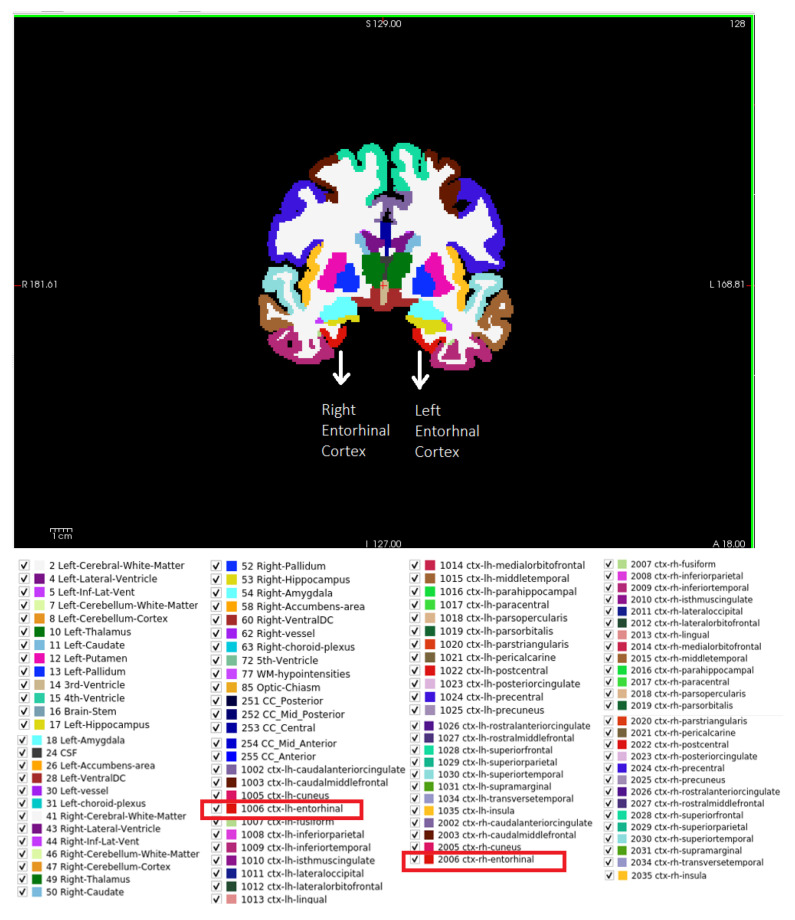
Brain regions in Desikan–Killiany atlas [43] presented by Freesurfer software. The entorhinal cortex area appears in red; code 1006 represents the left EC, and 2006 represents the right EC.

**Figure 5 sensors-23-05648-f005:**
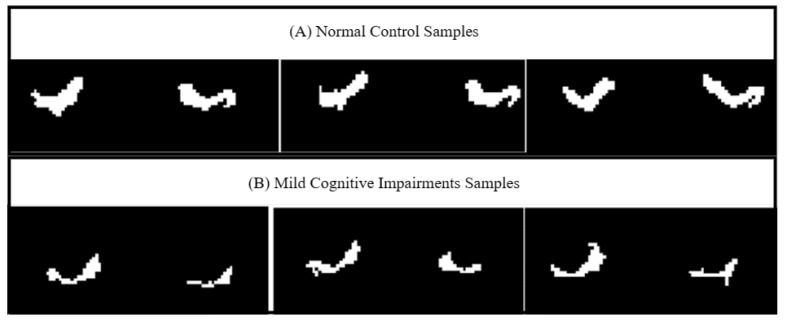
Extracted entorhinal cortex area for normal control and mild cognitive impairment in the current approach. (**A**) Normal area for EC compared to abnormal EC affected by MCI in (**B**); (**B**) shows the shrinkage of EC for MCI cases.

**Figure 6 sensors-23-05648-f006:**
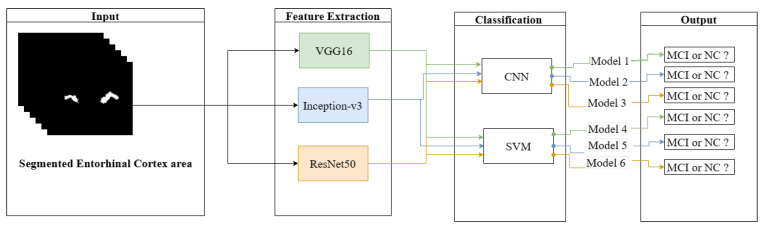
The classification approach used in this study.

**Figure 7 sensors-23-05648-f007:**
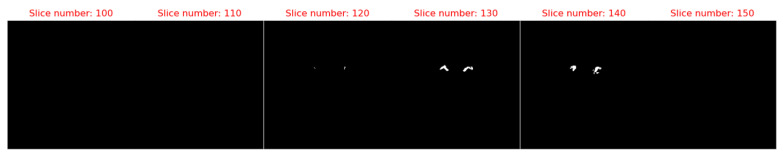
MRI slices along and including the EC area; the EC area is not observed in some MRI slices.

**Figure 8 sensors-23-05648-f008:**
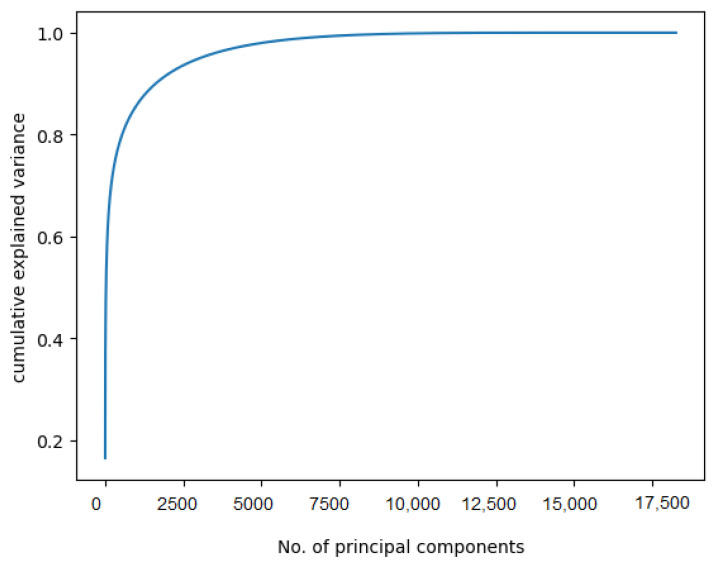
Scree plot for principal component analysis (PCA).

**Figure 9 sensors-23-05648-f009:**
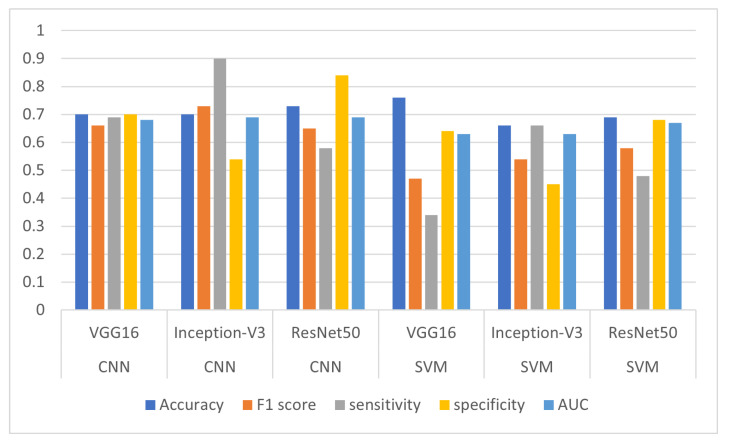
Results for accuracy, F1 score, sensitivity, specificity, and AUC metrics for the six experiments.

**Table 1 sensors-23-05648-t001:** Mild cognitive impairment classification systems based on whole brain image.

Reference	Input	Dataset	Architecture	Accuracy
[23] 2017	MRI	ADNI (MCI 400, NC 229)	CNN 9 layers	MCI vs. NC: 91%
[24] 2018	MRI	ADNI (MCI 193, NC 161)	CNN: dilated, residual, and dense connections	MCI vs. NC: 75%
[28] 2018	MRI	ADNI (MCI 70, NC 70)	CNN: autoencoder	MCI vs. NC: 86%
[26] 2019	MRI	ADNI (MCI 297, NC 315)	CNN: dense connections	MCI vs. NC: 98%
[27] 2020	MRI	ADNI (NC 162, MCIc 76)	CNN and ensemble learning	MCIc vs. NC: 79%
[22] 2021	MRI	ADNI (NC 85, EMCI 70, LMCI 70)	Transfer learning using VGG-19	EMCI vs. NC: 87% LMCI vs. NC: 89%

MCI: mild cognitive impairment; NC: normal control; EMCI: early MCI; LMCI: late MCI; CNN: convolutional neural networks.

**Table 2 sensors-23-05648-t002:** Mild cognitive impairment classification systems based on the hippocampus region.

Reference	Input	Dataset	Segmentation	Features Extraction	Classification	Accuracy MCI vs. NC
[30] 2017	MRI	ADNI (MCI 399, NC 228)	AAL Template	CNN	DL: CNN	66%
[31] 2018	MRI + DTI	ADNI (MCI 399, NC 228)	AAL Template	CNN	DL: CNN	80%
[33] 2019	MRI	ADNI (MCI 50, NC 50)	ICBM Template	GLCM and Gabor filters	ML: SVM	82%
[32] 2020	MRI	ADNI (MCI 60, NC 60)	3D slicer	CNN	DL: CNN	78.1%

ML: machine learning; DL: deep learning; AAL: automated anatomical labelling; ICBM: International Consortium for Brain Mapping; DTI: diffusion tensor imaging.

**Table 3 sensors-23-05648-t003:** Mild cognitive impairments classification systems based on at least the entorhinal cortex region.

Reference	Input	Dataset	ROI	Segmentation	Features	Classification	Performance
[36] 2019	MRI	ADNI + BU-ADC (NC 46, MCI 50)	EC, Hip, and other cortical measures	FreeSurfer	Volume	Logistic Regression	Accuracy NC vs. MCI: 94%
[35] 2020	MRI	ADNI (NC 194, MCI 200)	EC	FreeSurfer	Texture + Volume	Logistic Regression	AUCs NC vs. MCI: 71%
[37] 2021	MRI	MDCN (aMCI-s 29, aMCI-m 22, NC 26)	EC	FreeSurfer	Thickness, Surface Area, and Volume	Statistical Analyses: ANOVA	AUCs NC vs. aMCI-s: 76% NC vs. aMCI-m: 79%

ROI: region of interest; EC: entorhinal cortex; Hip: hippocampus; AUC: area under curve; aMCI-s: MCI single domain; aMCI-m: MCI multiple domains.

**Table 4 sensors-23-05648-t004:** Demographic characteristics (age and gender) of the participants included in the ADNI dataset; age is represented as mean ± standard deviation.

	MCI	NC
Sample Size	337	442
Age (year, mean ± SD)	76.3 ± 7.6	76.3 ± 6.3
Gender (male:female)	220:117	247:177

**Table 5 sensors-23-05648-t005:** Performance of the MCI classification system using different groups of MRI slices as inputs for the classification system; the feature extraction method and the classifier are fixed for the three experiments.

Experiment	MRI Slices Included	Feature Extraction	Classifier	Accuracy
Expt 1.1	Slice 0–255	VGG16	CNN	0.55
Expt 1.2	Slice 130–140	VGG16	CNN	0.58
Expt 1.3	Exclude uninformative slices	VGG16	CNN	0.60

**Table 6 sensors-23-05648-t006:** Classification accuracy based on the number of principal components (PCs).

Classifier	Number of PCs	Feature Extraction	Accuracy
SVM	2500	VGG16	0.45
SVM	5000	VGG16	0.51
SVM	7500	VGG16	0.53
SVM	10,000	VGG16	0.52
SVM	17,500	VGG16	0.52

**Table 7 sensors-23-05648-t007:** Accuracy of SVM classifier with tuned parameters.

Classifier	Feature Extraction	C	Degree	Gamma	Kernel	Accuracy
SVM	VGG16	0.1	2	scale	poly	0.561

**Table 17 sensors-23-05648-t017:** Results achieved using 5-fold cross-validation by CNN and SVM classifiers with VGG16, Inception-V3, and ResNet50 networks as feature extraction.

Experiment	Classifier	Feature Extraction	Accuracy	F1 Score	Sen	Spe	AUC
Expt 5.1	CNN	VGG16	0.70	0.66	0.69	0.70	0.68
Expt 5.2		Inception-V3	0.70	0.73	0.90	0.54	0.69
Expt 5.3		ResNet50	0.73	0.65	0.58	0.84	0.69
Expt 5.4	SVM	VGG16	0.76	0.47	0.34	0.64	0.63
Expt 5.5		Inception-V3	0.66	0.54	0.66	0.45	0.63
Expt 5.6		ResNet50	0.69	0.58	0.48	0.68	0.67

## Data Availability

The Alzheimer’s Disease Neuroimaging Initiative (ADNI) dataset used for the present study is publicly available at https://adni.loni.usc.edu/ (accessed on 20 April 2022).

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
