# Peer review of "An Optimized Deep Learning Model for Predicting Mild Cognitive Impairment Using Structural MRI"

_sensors, 2023, doi:10.3390/s23125648_

Round 1

Reviewer 1 Report

1. Highlight your contributions more precisely.

2. Add one paragraph to discuss a few recent non-deep learning-based works including the following work and discuss their limitations in the introduction section to present the importance of the deep learning methods.

[1] “An Efficient Blood-Cell Segmentation for the Detection of Hematological Disorders”, IEEE Transactions on Cybernetics, vol. 52, no. 10, pp. 10615-10626, 2022.

3. Add one paragraph to discuss more recent advancements in brain tumor detection including the following work. 

 [1] “Deep Convolutional Neural Network-based Automatic Detection of Brain Tumour,” In 2023 2nd International Conference for Innovation in Technology (INOCON), pp. 1-6, 2023.

4. Add one paragraph to discuss more details about ResNet as suggested in the following papers with giving proper citations.

 [1] “An efficient deep convolutional neural network based detection and classification of acute lymphoblastic leukemia”, Expert Systems with Applications, vol. 183, pp. 115311, 2021.

[2] “High Accuracy Hybrid CNN Classifiers for Breast Cancer Detection using Mammogram and Ultrasound Datasets,” Biomedical Signal Processing and Control, vol. 80, pp. 104292, 2023.

 [3] “An Efficient Detection and Classification of Acute Leukemia using Transfer Learning and Orthogonal Softmax Layer-based Model”, IEEE/ACM Transactions on Computational Biology and Bioinformatics, vol. 1, pp. 1-12, 2022.

[4] “A Systematic Review on Recent Advancements in Deep and Machine Learning based Detection and Classification of Acute Lymphoblastic Leukemia,” IEEE Access, 2022.

5. Add one paragraph to discuss SVM more clearly as suggested in the following papers with giving proper citations.

[1] “A lightweight deep learning system for automatic detection of blood cancer,” Measurement, pp. 110762, vol. 191, 2022.

   3. Highlight your contributions more precisely.

4. Highlight the limitations of existing works discussed in the literature (related work).

NA

Reviewer 2 Report

An Optimized Deep Learning Model for Predicting Mild Cognitive Impairment Using Structural MRI

1. Overall, the authors have made a good attempt. However, the authors; proposed method does not adequately describe their data. You should justify the effectiveness of the proposed technique theoretically. Hope this review will help you to improve further. 2. Section 1: Introduction: In my opinion, this Section describes the originality of the proposed issue. However, I suggest the authors to add further details about the advantages of the proposed approach with respect to the ones used commonly. 3. Justify the proposed method accuracy improved with recently published work 4. In the conclusion the authors should state the limitations, research findings and future works of their method.

Can be improved 

Reviewer 3 Report

The manuscript seems like a review article. The authors need to clarify why it is a research article using the word in the title and introduction. If it is a clinical study, the patient's consent or approved ID must be submitted from Institutional Review Board (IRB). If this work is like an artificial simulation and computer-based model study, it is unsuitable for this journal. Other comments are the following:

1) In the Introduction part, the second and third paragraphs could be merged into one paragraph with some linking words.

2) In Figure 1, the sign or any marking symbol can be used to assign the parts of gray matter (like in Figure 2). It will be beneficial for the general reader.

3) It needs to add the full form of TP, TN, FP, and FN in Equation 1.

4) The Conclusion part needs to be concisely described. More important findings/results can be included in this conclusion. The 'three different feature extraction neural networks and two classifiers…' is currently available finding in conclusion.

Reviewer 4 Report

The paper submitted by the authors provides helpful information to the literature. However, some corrections should be done before acceptance.

Specific comments 

In the abstract, please avoid the use of abbreviations in this section. Figure 1, the pictures included here do not clearly display the findings already mentioned. Figure 2, are you sure about the hippocampus? In my opinion, the ventral part of the hippocampus is not observed here. Line 95, please avoid the use of numbers. It is better to write “different reports”, and after you can include the numbers. Line 103, please rewrite the references as (13-15). Same line 111. Line 105, please add “table with parameters used in other works” Line 118, what’s MNI? Line 151, again, you must explain the abbreviation before. In figure 8, you should include the differences in the MCI cases. Lines 247-252, please clarify concerning the second version. Line 281, please confirm that you used 256 images for each, then could you provide the slice thickness? Line 457, curial? The conclusion section should be revised since it is very general. Here, the most relevant findings should be highlighted.

Round 2

Reviewer 2 Report

Accept

Reviewer 3 Report

The authors have made all the required corrections and replied appropriately to the reviewer's comments. The editor may accept the revised manuscript.